# Toward Early and Objective Hand Osteoarthritis Detection by Using EMG during Grasps

**DOI:** 10.3390/s23052413

**Published:** 2023-02-22

**Authors:** Néstor J. Jarque-Bou, Verónica Gracia-Ibáñez, Alba Roda-Sales, Vicente Bayarri-Porcar, Joaquín L. Sancho-Bru, Margarita Vergara

**Affiliations:** Department of Mechanical Engineering and Construction, Universitat Jaume I, E12071 Castellón, Spain

**Keywords:** hand function, hand osteoarthritis, electromyography, diagnosis, discriminant analysis

## Abstract

The early and objective detection of hand pathologies is a field that still requires more research. One of the main signs of hand osteoarthritis (HOA) is joint degeneration, which causes loss of strength, among other symptoms. HOA is usually diagnosed with imaging and radiography, but the disease is in an advanced stage when HOA is observable by these methods. Some authors suggest that muscle tissue changes seem to occur before joint degeneration. We propose recording muscular activity to look for indicators of these changes that might help in early diagnosis. Muscular activity is often measured using electromyography (EMG), which consists of recording electrical muscle activity. The aim of this study is to study whether different EMG characteristics (zero crossing, wavelength, mean absolute value, muscle activity) via collection of forearm and hand EMG signals are feasible alternatives to the existing methods of detecting HOA patients’ hand function. We used surface EMG to measure the electrical activity of the dominant hand’s forearm muscles with 22 healthy subjects and 20 HOA patients performing maximum force during six representative grasp types (the most commonly used in ADLs). The EMG characteristics were used to identify discriminant functions to detect HOA. The results show that forearm muscles are significantly affected by HOA in EMG terms, with very high success rates (between 93.3% and 100%) in the discriminant analyses, which suggest that EMG can be used as a preliminary step towards confirmation with current HOA diagnostic techniques. Digit flexors during cylindrical grasp, thumb muscles during oblique palmar grasp, and wrist extensors and radial deviators during the intermediate power–precision grasp are good candidates to help detect HOA.

## 1. Introduction

Hand osteoarthritis (HOA) is a chronic disease that may affect hand function. HOA can be found at different degrees in 81% of the elderly population [1,2], with a high prevalence especially in females aged over 50 years. HOA consequences are pain, joint deformity, and reduced hand mobility, strength and function [3,4]. Despite its high prevalence, HOA is a silent degenerating disorder that is clinically treated only in very severe situations. However, applying adequate treatments in early stages would benefit patient quality of life and could prevent disease progression [5].

HOA is usually diagnosed with a combination of different approaches, such as looking at risk factors, clinical presentations (e.g., nodes), radiographic images, laboratory results and subjective questionnaires [6]. Radiographic HOA is often diagnosed with the presence of osteophytes, loss of joint space, juxta articular sclerosis, local erosion and geodes, whereas clinical HOA is defined as the experience of joint pain, stiffness and discomfort [7]. However, symptoms often persist before HOA is observed via these methods [8]. Similarly, disability assessment in HOA is frequently performed using subjective questionnaires based on pain, satisfaction or physical hand function [9]. Therefore, patients’ diagnosis and follow-up very much depend on their willingness to recognize their functional limitations [10]. Very little attention has been paid to study forearm and hand muscles in individuals with HOA, perhaps because HOA is considered a problem of the joints. However, periarticular structures such as muscles, ligaments and synovial membranes may also be affected. Some studies have highlighted reduced muscle strength in patients with HOA [11,12]. Subjects diagnosed with HOA usually face increasing difficulty in performing simple handling tasks, reduced strength in lifting a ten pound weight and 10% less hand grasp strength [8]. Nunes et al. [4] found that HOA affects hand function and leads to functional deficits. However, none have studied whether the forearm muscles are significantly affected by HOA or differently used as a result of joint deterioration. Electromyographic (EMG) studies performed with knees have shown that strength deficits in the knee extensors of persons with osteoarthritis are partly due to the decreased recruitment of muscle fibers [13]. If muscle activation differs in the muscles around an osteoarthritic knee, then perhaps there are similar problems in the osteoarthritic hand’s forearm muscles. Surface EMG (sEMG) is a noninvasive technique that provides information on both the neural drive (amplitude) and temporal/phasic (shape) activation characteristics of muscles. In patients with osteoarthritis, Aspden [14] found that changes in muscle tissue seem to occur before joint degeneration and negatively affect joint stabilization. Brorsson et al. [15] studied the electromyography activity of extensor digitorum communis (EDC) and flexor carpi radialis (FCR) while female subjects with HOA performed functional activities to compare the results to a group of healthy subjects. They found statistically significant differences between the groups, finding that the HOA group used higher levels of muscle activation in daily tasks than the healthy group, and wrist extensors and flexors appeared to be equally affected. On the contrary, a recent work [16] compared the EMG signals of healthy individuals’ forearm muscles to those of HOA patients, and found an activation deficit of the wrist’s flexor and extensor muscles, even in initial HOA stages.

The merging of technology and medical science plays an essential role in the prevention, diagnosis and treatment of illnesses and diseases, including patient diagnostic data [17]. Health technology helps clinicians screen abnormalities and contributes to detecting clinical signs [18]. Thus, studying the forearm’s muscle signals while performing the most relevant grasps in daily life can lead to the finding of indicators that help detect HOA before the main symptoms appear. Given the large number of muscles that overlap in the forearm [19], it is practically impossible to isolate the surface EMG signal from each one. Therefore, in a previous work [20], we identified seven forearm areas with similar muscle activation patterns that can be used to characterize the forearm’s muscle activity while performing ADLs. However, such a study is hindered by the many EMG characteristics and their combinations that can be used to study muscle function. Selecting optimal EMG characteristics and the best combination between features and channels are challenging problems for accomplishing satisfactory classification performance [21,22]. In addition, an increment in EMG characteristics not only introduces redundancy into the function vector, but also increases complexity [21,23]. Of the existing characteristics, and besides muscle activation, new zero crossing (NZC), enhanced wavelength (EWL) and enhanced mean absolute value (EMAV) are those most frequently used in the literature for their efficiency and simplicity [24,25,26]. To date, no study has examined these EMG characteristics in an attempt to diagnose functional diseases such as HOA. Therefore, a study into the electromyography of forearm muscles (by considering the cited characteristics) would allow researchers to investigate whether subjects with HOA use different neuromuscular control compared to healthy subjects, especially in early disease stages.

One way to characterize the hand is studying hand grasp execution, which is composed mainly of two stages: the reach-to-object and grasp. The force needed to close a hand around and grasp an object is determined by several parameters, such as grasp stability (ability to resist external forces), and grasp security (resistance to slippery objects). Both depend on the grasp configuration [27,28], among other factors. Grasp configuration is determined by the type of applied grasp, and several grasp taxonomies have been reported in the literature in accordance with their purpose [29,30], such as the nine-type classification proposed in [30] for the commonest grasps used in activities of daily living (ADLs). This paper presents a study of the surface electromyography of forearm muscles (considering muscle activation, NZC, EWL and EMAV characteristics from seven representative forearm areas) while performing the commonest grasps used in day-to-day life with a twofold objective: (i) look for muscular forearm areas that are significantly affected or differently used by HOA in EMG terms; (ii) study if the affected EMG characteristics can be used as predictors to detect HOA in an early stage by using different combinations of them in discriminant analyses.

## 2. Materials and Methods

### 2.1. Experimental Study

Twenty HOA patients, all right-handed females (72 ± 9 years of age), and 22 right-handed healthy subjects (10 females and 12 males aged 32 ± 9 and 37 ± 11 years, respectively) were recruited for the experiment. All the subjects gave their written informed consent before participating in this study, which was approved by both the University and Hospital Ethics Committees (reference numbers CD/31/2019 and CD/27/2022). HOA patients were recruited by clinicians from among hospital patients showing different disease stages and levels of compromise, and none had undergone surgery. The recruitment was managed by our collaborator P. Granell in the framework of the collaboration agreement signed with the hospital. Healthy subjects were recruited among members of the research team, staff of the university and their relatives, and students, and inclusion criteria included subjects without a history of neuromuscular problems or injuries in the upper arm.

In a comfortable sitting posture, all the participants were asked to exert maximum effort without the help of other muscles other than those of the forearm and hand while performing six representative ADL grasps (Figure 1) based on the grasp taxonomy used in Vergara et al. [30], while recording muscular activity by means of sEMG: two-finger pad-to-pad pinch (P2D); cylindrical grasp (Cyl); lumbrical grasp (Lum); lateral pinch (LatP); oblique palmar grasp (Obl); and intermediate power–precision grasp (IntPP).

All the participants performed each grasp following an operator’s instructions: with their arm aligned with their trunk and an arm–forearm angle of 90°, the subject held a dynamometer by simulating the grasp to be analyzed without exerting force on it, and then exerted MGE for 2 s while maintaining the posture. Each MGE grasp was performed in a random order, with a 3-min break between each grasp to avoid muscle fatigue. For the normalization of sEMG signals, seven maximum voluntary contraction (MVC) records were measured with each subject (Figure 2): flexion and extension of the wrist, flexion and extension of fingers, ulnar and radial deviation of the wrist, and pronation of the forearm.

EMG signals were recorded with an 8-channel sEMG Biometrics Ltd. device at a sampling frequency of 1000 Hz. sEMG electrodes and dynamometer signals were synchronized by using the software provided by Biometrics. To place the sEMG electrodes, a grid was drawn on the forearm by using five easily identifiable anatomical landmarks, while the subject sat comfortably with their elbow resting on a table at an arm–forearm angle of 90º and the palm of their hand facing the subject. The grid defined 30 different spots covering the entire forearm surface (Figure 2). Following SENIAM recommendations [31], electrodes were placed longitudinally in the center of seven of these spots based on the spot groups obtained in a previous work [20] (Figure 3). Before placing electrodes, hair was removed by shaving and the skin was cleaned with alcohol.

### 2.2. Data Analysis

#### 2.2.1. Computed Parameters

Figure 4 shows the flowchart followed the data analysis. For efficiency and simplicity, those waveform characteristics most frequently used in the literature [21,23] were extracted (muscle activity, NZC, EWL and EMAV).

First of all, in order to define NZC, EWL and EMAV, the sEMG signals from the MGE records were filtered with a fourth-order bandpass filter between 25–500 Hz. Waveform characteristics (NZC, EWL, EMAV) were extracted from each record by considering the two seconds during which the maximum effort was made (according to the force signal recorded by the dynamometer). The proposed EMG characteristics were formulated according to [24,32], where *x* is the sEMG signal (mV), *L* is signal length and *T* is the selected threshold:(1)$$EWL=\overset{L}{\underset{i=2}{\sum }}\left|{\left(x_{i}-x_{i-1}\right)}^{p}\right|\;$$
(2)$$EMAV=\frac{1}{L}\overset{L}{\underset{i=1}{\sum }}\left|{\left(x_{i}\right)}^{p}\right|\;$$
$$where\;p=\left\{\begin{array}{c} 0.75,\;if\;i\geq 0.2L\;and\;i\leq 0.8L\;\\ 0.50,\;otherwise \end{array}\right.$$
(3)$$NZC=\left\{\begin{array}{c} 1,\;if\;x_{i}>T\;and\;x_{i+1}<T\\ \;or\;x_{i}<T\;and\;x_{i+1}>T\\ 0,\;otherwise \end{array}\right.;T=0\;$$

To determine muscle activity, the sEMG signals from the MGE records were filtered with a fourth-order bandpass filter between 25–500 Hz, rectified, filtered by a fourth-order low pass filter at 8 Hz and smoothed using Gaussian smoothing [33]. Later, they were normalized with the maximal values obtained in any of the seven MVC records measured with each subject. Finally, for each record, the average muscle activity recorded during the 2 s while performing maximum effort was computed for each spot (MA from this point onward).

#### 2.2.2. Global Description

First, as the HOA patients were all female, the gender effect was assessed among the four characteristics in the healthy subjects. For this purpose, the control group subjects were segregated by gender: subsample H_w (10 females) and subsample H_m (12 males). Then, a set of MANOVAs (one for each spot) was applied with the four characteristics as dependent variables, and with subsample and grasp type as factors. The MANOVAs compared subsamples H_w and H_m to assess the gender effect. For an overview of the results, the descriptive statistics (box-and-whisker plot) of all the characteristics (EWL, EMAV, NZC and MA values) per spot and grasp were computed for both subsamples H_w and H_m.

After checking the gender effect, a second set of MANOVAs (one per spot) was applied with the four characteristics as the dependent variable, and with sample (H_w and HOA) and grasp type as factors, as well as their interactions. For an overview of the results, the descriptive statistics (box-and-whisker plot) of all the characteristics (EWL, EMAV, NZC and MA values) per spot and grasp were computed for both samples H_w and HOA patients.

Finally, the four EMG characteristics were converted into 168 variables (4 EMG characteristics x 7 spots x 6 grasps). A MANOVA was performed with the EMG characteristics (168 variables) as dependent variables and sample (H_w and HOA) as the factor to identify which EMG characteristics, spots and grasps presented differences and which of them were, therefore, hindered by HOA.

#### 2.2.3. Can EMG Characteristics Be Used for Early HOA Diagnosis?

As a classification’s accuracy depends on the number and type of variables introduced into the model, 15 linear discriminant analyses (LDA) were performed (one for every possible combination of the four EMG characteristics; see Table 1) to locate a small set of predictive parameters to detect HOA. For each LDA, the EMG characteristics of spots and grasps that presented significant differences in the previous MANOVAs were taken as independent variables, and sample (HOA patient vs. H_w) was considered to be the grouping variable. Table 1 shows all the possible combinations of the EMG characteristics proposed in each LDA.

For LDAs, the stepwise method was used (predictors were entered sequentially), which searches for the highest correlated predictors. In particular, Wilks’ lambda was employed, which checks how well each independent variable (potential predictor) contributes to the model: 0 means total discrimination, and 1 denotes no discrimination. Each independent variable was tested by placing it in the model and then taking it out to generate a Λ statistic. The significance of change in Λ was measured using an F-test. The variable was entered in the model if the significance level of its F value was lower than the entry value (0.05), and it was removed if the significance level was higher than the removal value (0.1). Classification ability goodness was checked by a leave-one-out cross-validation, which repeats the analysis by taking one case out in each repetition. In addition, the percentage of correctly and incorrectly classified patients was checked.

## 3. Results

### 3.1. Are Forearm Muscles Significantly Affected or Differently Used by HOA in Terms of EMG Characteristics?

Appendix A presents the statistics (average and SD) of all the EMG characteristics for each spot, grasp and group. The next sections present that data in terms of box-and-whisker plots.

#### 3.1.1. Gender Effect in the Control Group Subjects

Figure 5 and Figure 6 show the box-and-whisker plots of the EMG characteristics segregated by gender and calculated for every grasp in each sample. As expected, the statistics shown in the box-and-whisker plots and the results of the first set of MANOVAs (Table 2) when comparing H_w and H_m found that gender significantly affected most of the EMG characteristics (*p* < 0.05), except for the ulnar deviators of the wrist (WR_UD and WE_UD). NZC was less affected by gender, and was affected only in FE and WE_RD. As gender affected the EMG characteristics, and to compare both target populations, from this point onward we only considered subsample H_w for the subsequent analyses as being representative of the control group.

#### 3.1.2. HOA Effect

Figure 7 and Figure 8 show the statistics of the EMG characteristics segregated by sample (HOA and H_w) and calculated per grasp in each sample by means of box-and-whisker plots. The results of the MANOVAs (Table 3) for comparing samples H_w and HOA show that group and grasp significantly affected most of the EMG characteristics (*p* < 0.05). Once again, NZC was that less affected by sample and its interaction with grasp.

Table 4 shows the results of the MANOVA (*p* < 0.05) performed to look for the EMG characteristics with significant differences between H_w and HOA. Regarding grasp types, Lum and IntPP were the grasps with the fewest significant variables in the different spots. On the contrary, Cyl and Obl were the grasps with the most significant variables. WF_UD, TM and WE_RD were the spots with the most significant variables, while WF_RD and DF were those with the least significant variables. Of the initial 168 variables (4 EMG characteristics × 7 spots × 6 grasps), 100 presented significant differences between samples. These 100 variables were used in the next LDAs.

### 3.2. Can EMG Characteristics Be Used for the Early Detection of HOA?

Table 5 shows the results of the discriminant analyses. The models in the table can be used to calculate discriminant scores F for each subject in such a way that when F is positive, the prediction is a healthy subject, and if F is negative, the subject has HOA. Superscripts ^i,j^ correspond to spot i, grasp j. The success ratio of the prediction using these discriminant scores ranged from 73.3% to 100%.

LDA1 had the worst success ratio, which was composed of only the NZC values. LDA2, LDA4, LAD5, LDA9, LDA10 and LDA14 had the highest success ratios (100%), with LDA4, LDA9, LDA10 and LDA14 requiring fewer characteristics and grasps with similar resulting models. Some LDAs obtained the same model, as can be observed in Table 5. LDA3 and LDA8 were the models with the fewest characteristics and required grasps (thumb muscles and Cyl grasp) and had a high success ratio (93.3%).

## 4. Discussion

In this work, an EMG study of forearm muscles (considering muscle activation, NZC, EWL and EMAV characteristics from seven representative forearm areas) while performing the commonest grasps of ADL was carried out with a twofold objective: 1) check if the EMG characteristics obtained from different forearm areas during grasps presented significant differences in HOA patients; 2) if these significant EMG characteristics can be used to help diagnose HOA.

### 4.1. Are Forearm Muscles Significantly Affected or Differently Used by HOA in EMG Terms?

First, and as expected, gender significantly affected the EMAV, MA and EWL characteristics, except for the ulnar deviators of the wrist (WR_UD and WE_UD). NZC was the least affected by gender, and was only affected in finger/wrist extensors and radial deviators (FE and WE_RD). Similarly, the vast majority of the EMG characteristics were also affected by condition (healthy women and HOA patients) and grasp type. In addition, NZC was once again the least affected. This seems reasonable because NZC is meant to approximate signal frequency, unlike EMAV, MA and EWL, which are related to signal amplitude and are, consequently, related more to grasping force, the decrease in which is a HOA symptom [34].

### 4.2. Can EMG Characteristics Be Used to Detect HOA Early?

From the LDA results, we observed that the EMG characteristics could help in detecting HOA. From all the tested combinations, six models presented the highest success ratio (100%), some of which presented similarities:
LDA2 and LDA5 were composed only of EWL values and required recording EMG signals from wrist flexors, ulnar deviators, thumb muscles, wrist extensors and radial deviators while performing all the grasps except the intermediate power–precision grasp. Not requiring MA characteristics would prevent MVC recordings and simplify the diagnosis method;LDA4, LDA9, LD10 and LDA14 required different combinations of EMG characteristics, but always from the same muscular forearm spots and grasps: digit flexors, thumb muscles, wrist extensors and radial deviators while performing the cylindrical, oblique-palmar grasp and intermediate power–precision grasp.

From the other models, LDA3 and LDA8 were seen to be the models with the fewest characteristics and required grasps (thumb muscles and Cyl grasp), and they also had a high success ratio (93.3%). This means that recording only one muscle spot while performing the cylindrical grasp could suffice to detect 93.3% of cases. Furthermore, not requiring MA characteristics would prevent MVC recordings and simplify the diagnosis method.

However, there were also differences in these models regarding the employed EMG characteristics:NZC values did not well-discriminate HOA patients;Muscle activity (MA) did not require any other characteristic to discriminate HOA patients, but required MVC recordings;EWL could very accurately discriminate, but needed information of more grasps;EMAV could very accurately discriminate, but always had to be accompanied by other EMG characteristics (MA or EWL).

There are few previous works that study different muscle activations in HOA, and comparisons with them must be made with caution, since the measurement protocols and the analyses performed are not the same. In [35], intrinsic muscles were considered in a fine manipulation activity, analyzing integrated activation as the only indicator and reaching the conclusion that although there were differences, when considering the longer execution time required by HOA patients, these differences disappeared. Despite [15] concluding that HOA patients require greater muscle activation for activities such as writing or cutting with scissors, [16] indicates that this activation is lower. However, although in both studies the signal was normalized, in [15], they do not indicate the application of any filtering. The novelty and importance of our work lies in considering different grasp types representative of ADLs; muscles whose activity is also representative of these ADLs; and indicators based not only on the amplitude of muscle activity, but also on the frequency domain of the signal. Therefore, ours is a broadening study in the pursuit of checking for differences in muscle activity due to HOA. The equations provided in this work show that the digit flexors during the cylindrical grasp, thumb muscles during the oblique palmar grasp and wrist extensors and radial deviators during the intermediate power–precision grasp were much more significant for detecting HOA than the other muscles and grasps. The recent work [16] found that early-stage HOA may contribute to the activation deficit of the flexor and extensor muscles of the wrist. The results herein reiterate wrist extensors, along with thumb muscles and digit flexors, as possible muscle indicators for detecting HOA.

## 5. Conclusions

This paper proposes using EMG characteristics to identify discriminant functions for the early detection of HOA. The discriminant results show very high success rates (between 93.3% and 100%), which suggests that EMG can be used as a preliminary step to confirm current HOA diagnostic techniques. In particular, digit flexors during the cylindrical grasp, thumb muscles during the oblique palmar grasp, and wrist extensors and radial deviators during the intermediate power–precision grasp are good candidates to help detect HOA. These results highlight the possibilities of merging technology and medical science as an essential role in the prevention, diagnosis and treatment of illnesses and diseases such as HOA. Furthermore, the results presented herein may help improve the control of hand prostheses and assistive exoskeletons, especially those intended for HOA patients. As limitations, note that there are other EMG parameters that have not been considered, that the sample of participants is limited both in number and degree of HOA compromise, and that we do not know what would happen with other pathologies (that could give similar indicators and be mislabeled as HOA). More studies are needed to check if these differences in EMG characteristics between healthy and HOA patients are present before strength loss in HOA.

## Figures and Tables

**Figure 1 sensors-23-02413-f001:**
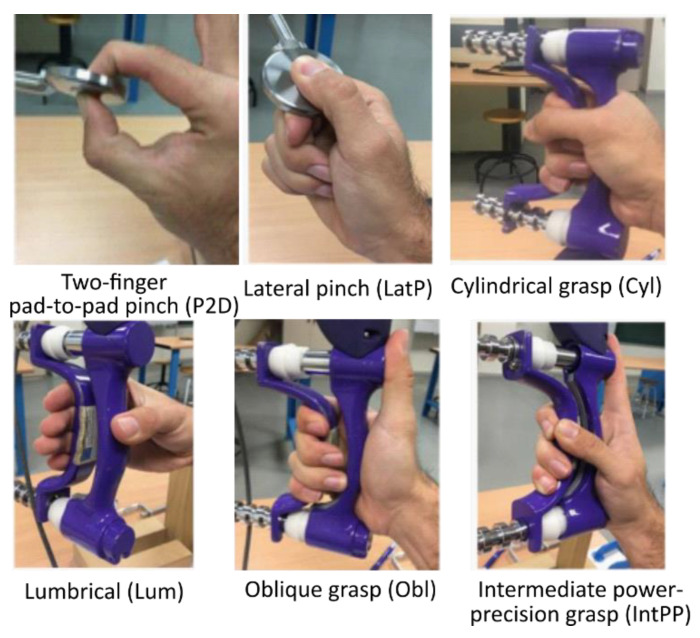
Six grasp types whose maximum grasping effort (MGE) was recorded. Grasp type definitions according to [30].

**Figure 2 sensors-23-02413-f002:**
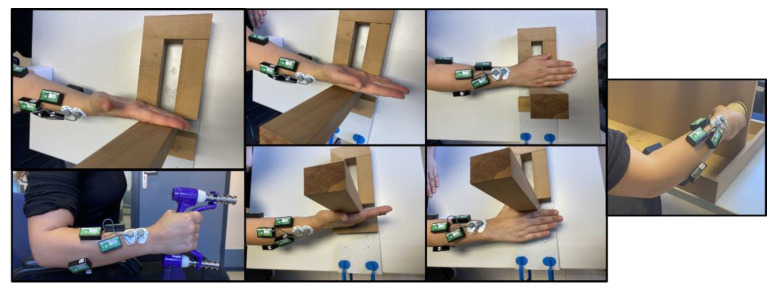
Seven MVC records for the normalization of the muscle activity signal. From left to right: flexion and extension of the fingers, flexion and extension of the wrist, ulnar and radial deviation of the wrist, and pronation of the forearm.

**Figure 3 sensors-23-02413-f003:**
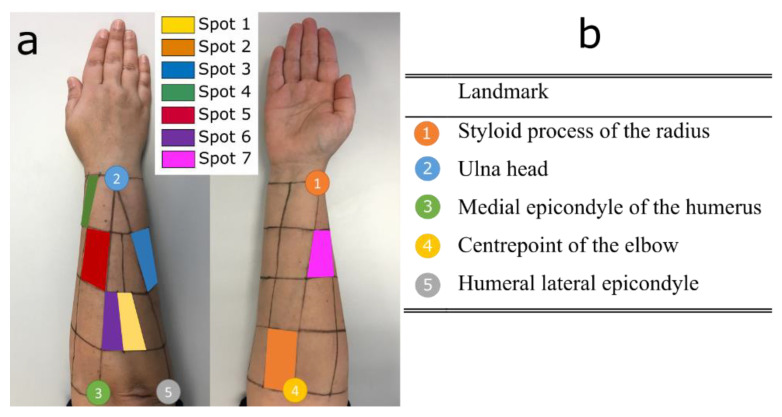
(**a**) Grid and spot areas selected for the sEMG recordings. (**b**) Five anatomical landmarks used to draw the grid. The signals from these seven spots are related to seven different movements according to [20]. Spot 1: wrist flexion and ulnar deviation (WF_UD); spot 2: wrist flexion and radial deviation (WF_RD); spot 3: digit flexion (DF); spot 4: thumb extension and abduction/adduction (TM); spot 5: finger extension (FE); spot 6: wrist extension and ulnar deviation (WE_UD); spot 7: wrist extension and radial deviation (WE_RD).

**Figure 4 sensors-23-02413-f004:**
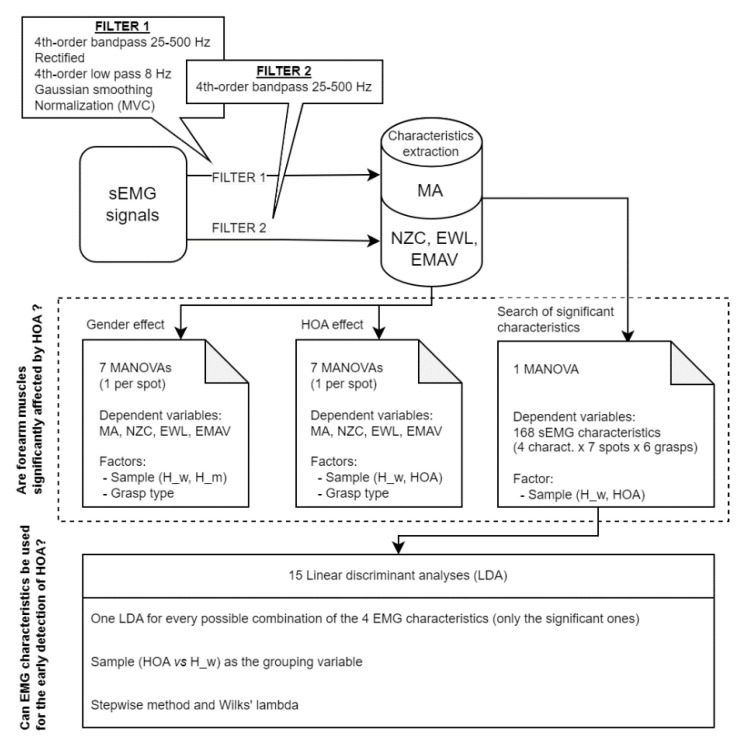
Flowchart of the methodology followed in this paper.

**Figure 5 sensors-23-02413-f005:**
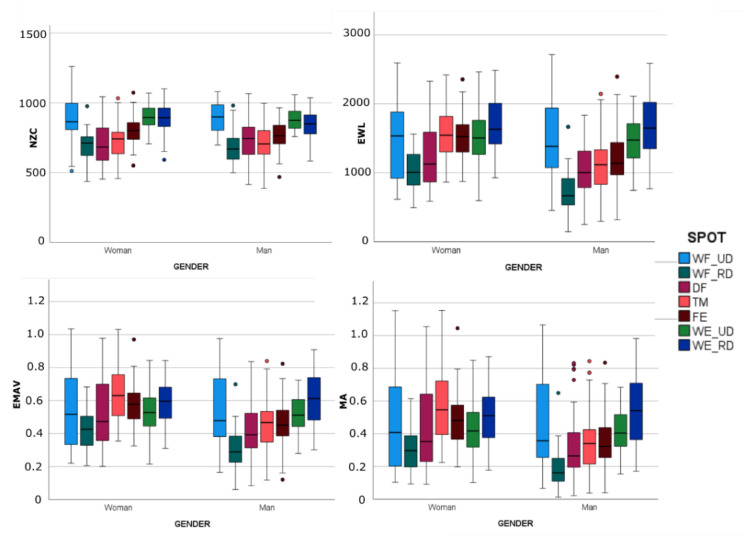
Box–and-whisker plots (horizontal central mark in the boxes is the median; the edges of the boxes are the 25th and 75th percentiles; whiskers extend to 1.5 times the interquartile range and outliers are marked as color circles) of the EMG characteristics segregated by gender and calculated per spot in each sample. Wrist flexion and ulnar deviation (WF_UD); wrist flexion and radial deviation (WF_RD); digit flexion (DF); thumb extension and abduction/adduction (TM); finger extension (FE); wrist extension and ulnar deviation (WE_UD); wrist extension and radial deviation (WE_RD).

**Figure 6 sensors-23-02413-f006:**
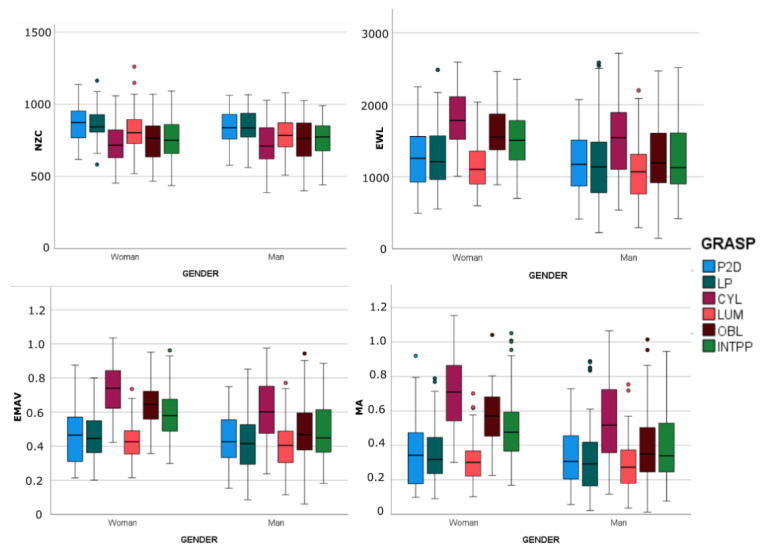
Box-and-whisker plots (horizontal central mark in the boxes is the median; the edges of the boxes are the 25th and 75th percentiles; whiskers extend to 1.5 times the interquartile range and outliers are marked as color circles) of the EMG characteristics segregated by gender and calculated per grasp in each sample.

**Figure 7 sensors-23-02413-f007:**
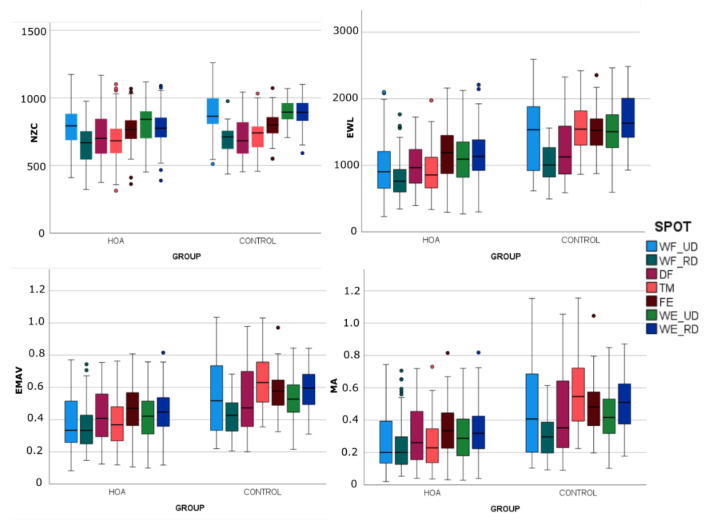
Box-and-whisker plots (horizontal central mark in the boxes is the median; the edges of the boxes are the 25th and 75th percentiles; whiskers extend to 1.5 times the interquartile range and outliers are marked as color circles) of the EMG characteristics segregated by group and calculated per spot in each sample. Abbreviations are defined the Figure 5 caption.

**Figure 8 sensors-23-02413-f008:**
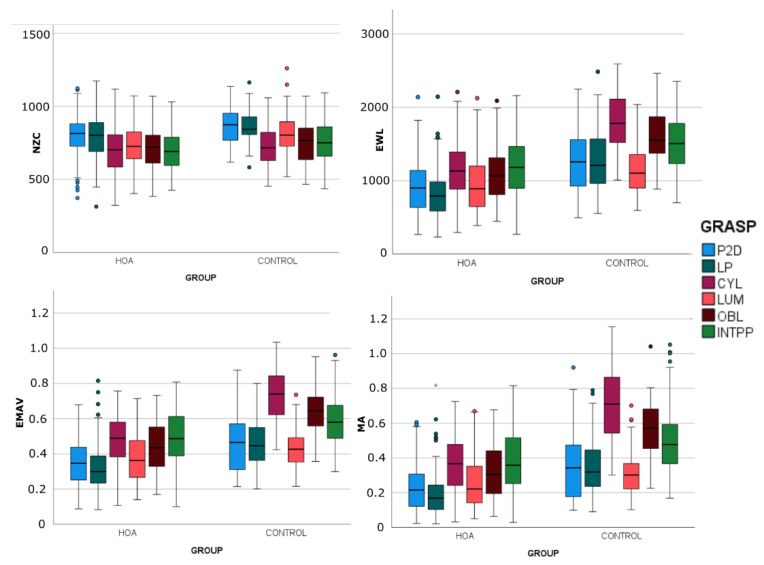
Box-and-whisker plots (horizontal central marks in the boxes correspond to the median; the edges of the boxes are the 25th and 75th percentiles; whiskers extend to 1.5 times the interquartile range and outliers are marked as color circles) of the EMG characteristics segregated by group and calculated per grasp in each sample.

**Table 1 sensors-23-02413-t001:** All the different combinations of the EMG characteristics of all the performed LDAs.

	NZC	EWL	EMAV	MA
LDA1	X			
LDA2		X		
LDA3			X	
LDA4				X
LDA5	X	X		
LDA6	X		X	
LDA7	X			X
LDA8		X	X	
LDA9		X		X
LDA10			X	X
LDA11	X	X	X	
LDA12	X		X	X
LDA13	X	X		X
LDA14		X	X	X
LDA15	X	X	X	X

**Table 2 sensors-23-02413-t002:** Results in columns of the set of MANOVAs. The EMG characteristics that significantly differed between H_w and H_m are indicated. Abbreviations are defined the Figure 5 caption.

	Spot						
Factor	WF_UD	WF_RD	DF	TM	FE	WE_UD	WE_RD
Gender		EWL EMAV MA	EWL EMAV MA	EWL EMAV MA	NZC EWL EMAV MA		NZC
Grasp type	NZC EWL EMAV MA	NZC EWL EMAV MA	NZC EWL EMAV MA	NZC EWL EMAV MA	NZC EWL EMAV MA	EWL EMAV MA	NZC EWL EMAV MA
Interaction			EWL EMAV MA				

**Table 3 sensors-23-02413-t003:** Results in columns of the set of MANOVAs. The EMG characteristics that significantly differ between the H_w and HOA patients samples are indicated. Abbreviations are defined in the Figure 5 caption.

	Spot						
Factor	WF_UD	WF_RD	DF	TM	FE	WE_UD	WE_RD
Sample	NZC EWL EMAV MA	EWL EMAV MA	EWL EMAV MA	EWL EMAV MA	NZC EWL EMAV MA	NZC EWL EMAV MA	NZC EWL EMAV MA
Grasp type	NZC EWL EMAV MA	NZC EWL EMAV MA	NZC EWL EMAV MA	NZC EWL EMAV MA	NZC EMAV MA	EWL EMAV MA	NZC EWL EMAV MA
Interaction	EWL EMAV MA	EWL EMAV MA	EWL EMAV MA	EWL EMAV MA	MA		EMAV MA

**Table 4 sensors-23-02413-t004:** Results of the MANOVA with the combined variable grasp x spot x EMG characteristic as input. Variables that significantly differ between H_w and HOA patients depend on the spot and grasp type. Abbreviations are defined in the Figure 5 caption.

				Spot			
Grasp Type	WF_UD	WF_RD	DF	TM	FE	WE_UD	WE_RD
P2D	EWL	EWL EMAV MA		EWL EMAV MA	EWL EMAV MA	EWL EMAV MA	NZC EWL EMAV MA
LatP	EWL EMAV MA	EWL EMAV MA		NZC EWL EMAV MA	NZC EWL EMAV MA	EWL EMAV MA	NZC EWL EMAV MA
CyL	EWL EMAV MA	EWL EMAV MA	EWL EMAV MA	EWL EMAV MA	EWL EMAV MA	EWL EMAV MA	EWL EMAV MA
Lum	EWL			EWL EMAV MA		NZC	NZC EWL
Obl	EWL EMAV MA	EWL EMAV	EWL EMAV MA	EWL EMAV MA	EWL EMAV MA	NZC EWL EMAV MA	NZC EWL EMAV MA
IntPP	EWL EMAV MA			EWL EMAV MA	NZC	NZC	NZC EWL EMAV MA

**Table 5 sensors-23-02413-t005:** The success ratios and models obtained from the different performed LDAs.

	Success Ratio	Model
LDA1	73.3%	0.013·NZC^WE_RD,Lum^ -10.383
LDA2 & LDA5	100%	0.002·EWL^WF_UD,Obl^ + 0.003·EWL^TM,LatP^ + 0.004·EWL^TM,Cyl^ − 0.004·EWL^TM,Lum^ − 0.002·EWL^WE_RD,P2D^-4.198
LDA3 & LDA8	93.3%	8.065·EMAV^TM,Cyl^ − 4.399
LDA4	100%	3.163·MA^DF,Obl^ + 8.121·MA^TM,Cyl^ − 4.986·MA^WE_RD,IntPP^ − 3.232
LDA6 & LDA11	93.3%	7.902·EMAV^TM,Cyl^ + 0.005·NZC^WE_RD,IntPP^ − 8.483
LDA7	93.3%	6.514·MA^TM,Cyl^ + 0.006·NZC^WE_RD,IntPP^ − 7.512
LDA9	100%	0.002·EWL^DF,Obl^ + 8.542·MA^TM,Cyl^ − 5.566·MA^WE_RD,IntPP^ − 4.140
LDA10	100%	3.277·MA^DF,Obl^ + 8.215 MA^TM,Cyl^ − 6.313·EMAV^WE_RD,IntPP^ − 2.127
LDA12, LDA13 & LDA15	93.3%	6.514·MA^TM,Cyl^ + 0.006·NZC^WE_RD,IntPP^ − 7.512
LDA14	100%	0.002·EWL^DF,Obl^ + 8.681·MA^TM,Cyl^ − 7.112·EMAV^WE_RD,IntPP^ − 2.938

## Data Availability

The data presented in this study are available upon request from the corresponding author.

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
