# Peer review of "Toward Early and Objective Hand Osteoarthritis Detection by Using EMG during Grasps"

_sensors, 2023, doi:10.3390/s23052413_

Round 1
Reviewer 1 Report
This article mainly studies the characteristics of forearm myoelectric signals under six grasping postures, discusses the feasibility of identifying grasping postures through these characteristics, and makes a comparative study of the differences between forearm myoelectric signals of patients and normal people. There are several issues that need further discussion, as follows: 1. What is the purpose of 1 to 5 identified in Figure 2? Are these numbers (1 to 5) used to determine the placement of myoelectric electrodes on the forearm? 2. Does MA in Table 1 mean multiple activity? How can MA express it with mathematical formula? In this section (2.2.1. Computed parameters), MA does not provide specific mathematical formulas. If not, it is not easy to understand the following contents (Table 2, Table 3 and Table 4), 3. Is there any physical difference between LDA1 and LDA 15 in Table 1? Why 15 linear discriminant analysis, can this number be more or less? If it is a permutation and combination of four different factors (NZC, EWL, EMAV, MA), and one has 24 different combinations, is it necessary to make 24 linear discriminant analysis?Author Response
This article mainly studies the characteristics of forearm myoelectric signals under six grasping postures, discusses the feasibility of identifying grasping postures through these characteristics, and makes a comparative study of the differences between forearm myoelectric signals of patients and normal people. There are several issues that need further discussion, as follows:
- What is the purpose of 1 to 5 identified in Figure 2? Are these numbers (1 to 5) used to determine the placement of myoelectric electrodes on the forearm?
ANSWER: These numbers were used as landmarks in order to draw the grid. Figure 2 and its caption have been modified to make it clearer. - Does MA in Table 1 mean multiple activity? How can MA express it with mathematical formula? In this section (2.2.1. Computed parameters), MA does not provide specific mathematical formulas. If not, it is not easy to understand the following contents (Table 2, Table 3 and Table 4),
ANSWER: MA is just the average of the muscle activity during the 2 seconds performing the maximum effort. There is not a formula in the manuscript, but it is clearly explained in the last paragraph of section 2.2.1: ’To determine muscle activity, the sEMG signals from the MGE records were filtered with a fourth-order bandpass filter between 25-500 Hz, rectified, filtered by a fourth-order low pass filter at 8 Hz and smoothed by Gaussian smoothing [33]. Later they were normalized with the maximal values obtained in any of the seven MVC records measured with each subject. Finally, for each record, the average muscle activity recorded during the 2 seconds while performing maximum effort was computed for each spot (MA from this point onward)’. Now, we have added a flowchart in the manuscript for a better understanding of the methodology used. - Is there any physical difference between LDA1 and LDA 15 in Table 1? Why 15 linear discriminant analysis, can this number be more or less? If it is a permutation and combination of four different factors (NZC, EWL, EMAV, MA), and one has 24 different combinations, is it necessary to make 24 linear discriminant analysis?
ANSWER: The number of possible combinations of the 4 parameters (NZC, EWL, EMAV, MA) is
.
Table 1 lists all the possible combinations. Input parameters for LDA1 are the NZC of those spots and grasps that presented significant differences in the previous ANOVA (Table 4), while LDA15 uses all the 4 parameters with significant differences. Now, it has been clarified in the manuscript.
Reviewer 2 Report
Authors propose an early and objective hand osteoarthritis detection by using EMG signals during grasps. The aim of this research is to study whether different EMG feature extraction approaches are feasible alternatives for using forearm and hand EMG signals to detect HOA patients’ hand function. sEMG is used to measure the electrical activity of the dominant hand’s forearm muscles with 22 healthy subjects and 20 HOA patients, while performing maximum force during six representative grasp types.
This is a good paper, however, I have some comments and remarks.
- Abstract needs to be rewritten to understand the summary of the work done highlighting the principal contributions.
- In the introduction, the comparison with previous works must be more precise in order to highlight the real contribution of this work. In addition, the motivation and background of wide practical use of the theoretic results presented should be clearly emphasized to facilitate the readers.
- English is generally good, but needs to be polished further. The manuscript should be formatted better and some spelling and grammar should be checked carefully.
- The selection of the related useful feature extraction approaches should be more discussed. How the authors chose these approaches only based on the literature? I think that it's better to test some features and select the more suitable approaches.
- Regarding the feature extraction approaches, Authors use a mixed time-domain and frequency domain features, this issue should be more addressed in the paper. Why authors use a rectified-smoothed EMG (lines 171-172) for frequency processing (NZC) ?
- Muscle selection and electrode placement are well detailed and justified.
- The selection of the sampling frequency should be justified (Shannon theory). I think that 1Khz is too low for accurate signal reconstruction.
-What kind of filter has been used (IIR?) for EMG preprocessing ?
- Discussion part is too reduced, this can be extended by adding more details.
- The Conclusion should be rewritten by integrating the limitations and the perspective.
Concluding, the paper has potential to be appreciated by the readers and the above comments are formulated such that to enhance its impact.
Author Response
Authors propose an early and objective hand osteoarthritis detection by using EMG signals during grasps. The aim of this research is to study whether different EMG feature extraction approaches are feasible alternatives for using forearm and hand EMG signals to detect HOA patients’ hand function. sEMG is used to measure the electrical activity of the dominant hand’s forearm muscles with 22 healthy subjects and 20 HOA patients, while performing maximum force during six representative grasp types.
This is a good paper, however, I have some comments and remarks.
ANSWER: Thanks to the reviewer for these comments. To better clarify the novelty of the paper, the abstract, discussion and conclusion have been rephrased highlighting the novel methodological aspects and results.
- Abstract needs to be rewritten to understand the summary of the work done highlighting the principal contributions.
ANSWER: The abstract has been modified, highlighting the principal contributions as suggested by the reviewer.
- In the introduction, the comparison with previous works must be more precise in order to highlight the real contribution of this work. In addition, the motivation and background of wide practical use of the theoretic results presented should be clearly emphasized to facilitate the readers.
ANSWER: Now, the introduction has been rephrased. We have better emphasized the motivation and background, highlighting the real contributions of this work: 1) look for muscular forearm areas that are significantly affected by HOA in EMG terms; 2) study whether the affected EMG characteristics can be used as predictors to detect HOA in an early stage by using different combinations of them in discriminant analyses
- English is generally good, but needs to be polished further. The manuscript should be formatted better and some spelling and grammar should be checked carefully.
ANSWER: Ok. English has been carefully checked by a professional native English proofreader.
- The selection of the related useful feature extraction approaches should be more discussed. How the authors chose these approaches only based on the literature? I think that it's better to test some features and select the more suitable approaches.
ANSWER: Due to the large number of features that can be extracted, a bibliographic research was carried out to identify those parameters that provided best classification results. From these parameters, we considered the simplest ones in order to reduce computational cost. We have detailed the procedure in the new version of the manuscript: “For efficiency, those waveform characteristics most frequently used in the literature [25,26] and implying less computational cost were extracted (Muscle activity, NZC, EWL and EMAV).”
- Regarding the feature extraction approaches, Authors use a mixed time-domain and frequency domain features, this issue should be more addressed in the paper. Why authors use a rectified-smoothed EMG (lines 171-172) for frequency processing (NZC)?
ANSWER: We believe that the reviewer has not fully understood the procedure. Rectified-smoothed EMG was only obtained to define the muscle activity. The other EMG characteristics (NZC, EWL and EMAV) were obtained from the original signal, only filtered with a fourth-order bandpass filter between 25-500 Hz, as commonly done in the literature. Now, we have added some text to clarify feature extraction: “First of all, in order to determine NZC, EWL and EMAV, the sEMG signals from the MGE records were only filtered with a fourth-order bandpass filter between 25-500 Hz. To determine muscle activity, the sEMG signals from the MGE records were filtered with a fourth-order bandpass filter between 25-500 Hz, rectified, filtered by a fourth-order low pass filter at 8 Hz and smoothed by Gaussian smoothing [30].”
- Muscle selection and electrode placement are well detailed and justified.
ANSWER: Ok.
- The selection of the sampling frequency should be justified (Shannon theory). I think that 1Khz is too low for accurate signal reconstruction.
ANSWER: Although signal reconstruction needs higher sampling frequency, this is not our aim. As indicated in the literature [1,2], 1Khz is enough for the extraction of EMG features for pattern recognition, which is our aim.
[1] Fang, C.; He, B.; Wang, Y.; Cao, J.; Gao, S. EMG-Centered Multisensory Based Technologies for Pattern Recognition in Rehabilitation: State of the Art and Challenges. Biosensors 2020, 10, 85. https://doi.org/10.3390/bios10080085
[2] A. Saikia, S. Mazumdar, N. Sahai, S. Paul and D. Bhatia, "Comparative study and feature extraction of the muscle activity patterns in healthy subjects," 2016 3rd International Conference on Signal Processing and Integrated Networks (SPIN), Noida, India, 2016, p. 147-151, doi: 10.1109/SPIN.2016.7566678.
-What kind of filter has been used (IIR?) for EMG preprocessing ?
ANSWER: We have made some changes in the manuscript, and have included a flowchart to clarify that the sEMG records were filtered with two different set of filters:
- For the extraction of the NZC, EWL and EMAV, sEMG signals from the MGE records were filtered with a fourth-order bandpass filter between 25-500 Hz.
- For the extraction of MA, signals were filtered with a 4th-order bandpass filter between 25-500 Hz, rectified, filtered by a 4th-order low pass filter at 8 Hz, and smoothed by Gaussian smoothing.
- Discussion part is too reduced, this can be extended by adding more details.
ANSWER: Discussion part has been extended by adding more details of the results and comparing them with previous works.
- The Conclusion should be rewritten by integrating the limitations and the perspective.
ANSWER: Conclusion has been extended by integrating limitations and future works: As limitations, note that there are other EMG parameters that have not been considered, that the sample of participants is limited both in number and in degrees of HOA compromise), as well as that we do not know what would happen with other pathologies (that could give similar indicators and be mislabeled as HOA). More studies are needed to check if these differences in EMG characteristics between healthy and HOA patients are present before strength loss in HOA.
Concluding, the paper has potential to be appreciated by the readers and the above comments are formulated such that to enhance its impact.
Reviewer 3 Report
The manuscript “Towards early and objective hand osteoarthritis detection by using EMG during grasps” aimed to investigate if electromyographic analyses are an useful tool to diagnose hand osteoarthritis at its onset. I commend the authors for their work in conducting the experiment and preparing the manuscript. General comments and specific points and sections are provided below:
The research question is interesting and based off previous observation.
Please state the hypothesis of the study – I suppose it was defined a priori
The methods are adequate to respond to the study’s question. However, an important aspect that has not ben stated is the blinding of the examinators and analyzers. Were investigator/people who analyzed the data aware of the groups (i.e., HOA vs healthy) the participants were part of? This could incur in potential biases. If this was the case, please state this as a limitation of the study.
Results are clearly presented and described. I commend the authors for their nice work in making their results clear.
Although brief, the discussion is well organized and comprehensive. The authors avoid speculation to the maximum, which is commendable.
The conclusion is concise and based off the obtained results, as it should be.
Author Response
The manuscript “Towards early and objective hand osteoarthritis detection by using EMG during grasps” aimed to investigate if electromyographic analyses are an useful tool to diagnose hand osteoarthritis at its onset. I commend the authors for their work in conducting the experiment and preparing the manuscript. General comments and specific points and sections are provided below:
The research question is interesting and based off previous observation.
Please state the hypothesis of the study – I suppose it was defined a priori
ANSWER: Now, the introduction has been rephrased to better emphasized the motivation and hypothesis of the study, highlighting the real contributions of this work: 1) look for muscular forearm areas that are significantly affected by HOA in EMG terms; 2) study whether the affected EMG characteristics can be used as predictors to detect HOA in an early stage by using different combinations of them in discriminant analyses.
The methods are adequate to respond to the study’s question. However, an important aspect that has not ben stated is the blinding of the examinators and analyzers. Were investigator/people who analyzed the data aware of the groups (i.e., HOA vs healthy) the participants were part of? This could incur in potential biases. If this was the case, please state this as a limitation of the study.
ANSWER: It was impossible for the operators not to perceive the deformation of the hands when instrumenting them, so that the operators knew if the participant was healthy or HOA patient. Even so, the operators were always the same and gave the same instructions to all participants, and used the same equipment, so that no bias is expected during the data recording. The analyses performed required labelling the data with the condition (healthy vs HOA), but again, the analyzers can’t introduce any bias in the results from the analyses performed.
Results are clearly presented and described. I commend the authors for their nice work in making their results clear.
Although brief, the discussion is well organized and comprehensive. The authors avoid speculation to the maximum, which is commendable.
The conclusion is concise and based off the obtained results, as it should be.
ANSWER: Thank you for your positive comments.
Reviewer 4 Report
Paper is important for health informatics. Following revisions are to be incorporated before publication-
(1) How data sets are established for doing this research work?
(2) The quality of figures is not appropriate. Author should modify.
(3) Add more results in the form of some graphs and tables.
(4) What are the strong features of this research work? Author must explain.
(5) How the parameters for simulations are selected?
(6) All tables and figures should be explained clearly.
(7) The methodology of the paper should be clearly explained with appropriate flow charts.
(8) Highlight the more applications of the proposed technique.
(9) What are the major issues in the proposed work?
(10) Author must add following papers-
(a) Real-Time Classification of EMG Myo Armband Data Using Support Vector Machine
(b) A Real-Time Capable Linear Time Classifier Scheme for Anticipated Hand Movements Recognition from Amputee Subjects Using Surface EMG Signals
(c) An Improved SSA-Based Technique for EMG Removal from ECG
(d) A real time electromyostimulator linked with EMG analysis device
(e) Effect of decimation on the classification rate of non-linear analysis methods applied to uterine EMG signals
(f) Parameters extraction and monitoring in uterine EMG signals. Detection of preterm deliveries
(g) Flexible Analytic Wavelet Transform Based Features for Physical Action Identification Using sEMG Signals
(h) Removal of ECG interference from surface respiratory electromyography
(i) Power line interference rejection from surface electromyography signal using an adaptive algorithm
(j) Multi-Feature Fusion Method for Identifying Carotid Artery Vulnerable Plaque
(k) 3D Coronary Artery Reconstruction by 2D Motion Compensation Based on Mutual Information
(l) Robust retinal blood vessel segmentation using convolutional neural network and support vector machine
(m) Real-time estimation of hospital discharge using fuzzy radial basis function network and electronic health record data
(n) An efficient ALO-based ensemble classification algorithm for medical big data processing
(o) Multiscale Graph Cuts Based Method for Coronary Artery Segmentation in Angiograms
(p) Changes in scale-invariance property of electrocardiogram as a predictor of hypertension
(q) Assessment of qualitative and quantitative features in coronary artery MRA
(r) A frugal and innovative telemedicine approach for rural India – automated doctor machine
(s) Study of murmurs and their impact on the heart variability
(t) Coronary three-vessel disease with occlusion of the right coronary artery: What are the most important factors that determine the right territory perfusion?
Author Response
Paper is important for health informatics. Following revisions are to be incorporated before publication-
(1) How data sets are established for doing this research work?
ANSWER: The question is pertinent, especially regarding the HOA data set. HOA patients were recruited by clinicians from among hospital patients showing different disease stages and levels of compromise, and none had undergone surgery. The recruitment was managed by our collaborator Pablo Granell, in the framework of the collaboration agreement signed with the hospital of Castellón de la Plana. Healthy subjects were recruited among members of the research team, staff of the University, relatives and students, and inclusion criteria included subjects without history of neuromuscular problems or injuries in the upper arm. This information has been added to the manuscript.
(2) The quality of figures is not appropriate. Author should modify.
ANSWER: The quality of figures has been modified. In addition, we have increased the size of the numbers in the figures to make them easier to read.
(3) Add more results in the form of some graphs and tables.
ANSWER: We have added detailed results as supplementary material: average and SD values of the four characteristics for each spot, grasp and group (healthy women, healthy men and HOA) and it has indicated in the manuscript: “Table I of supplementary material presents the statistics (average and SD) of all the EMG characteristics for each spot, grasp and group. The next sections present that data in terms box and whiskers plots”.
(4) What are the strong features of this research work? Author must explain.
ANSWER: Now, in the introduction section, we have better emphasized the motivation and background, highlighting the real contributions of this work: 1) look for muscular forearm areas that are significantly affected by HOA in EMG terms; 2) study whether the affected EMG characteristics can be used as predictors to detect HOA in an early stage by using different combinations of them in discriminant analyses.
(5) How the parameters for simulations are selected?
ANSWER: We are not sure about this question, since we are not doing simulations. We believe that the reviewer is concerned with the choice of which parameters we extract from the EMG signals. If it is the case and Due to the large number of features that can be extracted, a bibliographic research was carried out to identify those parameters that provided best classification results. From these parameters, we considered the simplest ones in order to reduce computational cost. We have detailed the procedure in the new version of the manuscript: “For efficiency, those waveform characteristics most frequently used in the literature [25,26] and implying less computational cost were extracted (Muscle activity, NZC, EWL and EMAV).”
(6) All tables and figures should be explained clearly.
ANSWER: Captions of tables and figures have been improved to make them clearer.
(7) The methodology of the paper should be clearly explained with appropriate flow charts.
ANSWER: Ok. Now, the methodology of the paper has been better explained with a flowchart.
(8) Highlight the more applications of the proposed technique.
ANSWER: More applications of the results have been added: “Furthermore, the results presented herein may help to improve the control of hand prostheses and assistive exoskeletons, especially those intended for HOA patients.”
(9) What are the major issues in the proposed work?
ANSWER: Limitations and future works have been detailed now in the conclusion section: As limitations, note that there are other EMG parameters that have not been considered, that the sample of participants is limited both in number and in degrees of HOA compromise), as well as that we do not know what would happen with other pathologies (that could give similar indicators and be mislabeled as HOA). More studies are needed to check if these differences in EMG characteristics between healthy and HOA patients are present before strength loss in HOA.
(10) Author must add following papers-
ANSWER: We have carefully revised these papers, and the relevant ones have been added (the three papers listed below). The other papers are not related with our work, since they deal about coronary artery problems or uterine signals and do not use feature extraction techniques.
(a) Real-Time Classification of EMG Myo Armband Data Using Support Vector Machine
(b)A Real-Time Capable Linear Time Classifier Scheme for Anticipated Hand Movements Recognition from Amputee Subjects Using Surface EMG Signals
(g) Flexible Analytic Wavelet Transform Based Features for Physical Action Identification Using sEMG Signals
Reviewer 5 Report
This paper presents a significant study on hand osteoarthritis detection by using EMG during grasps. Materials and methods are given in detail. Some issues can be improved.
1) Section 2.1: "All the subjects gave their written informed consent before participating in this study, which was approved by both the University and Hospital Ethics Committees". It would be better if the authors provide the evidence or certificate, etc.
2) Motivations and new contributions/originality should be mentioned in introduction.
3) Section 2.1: Experiments should be detailed.
4) Quality of figures: Numbers in pictures are quite smalll.
Author Response
This paper presents a significant study on hand osteoarthritis detection by using EMG during grasps. Materials and methods are given in detail. Some issues can be improved.
1) Section 2.1: "All the subjects gave their written informed consent before participating in this study, which was approved by both the University and Hospital Ethics Committees". It would be better if the authors provide the evidence or certificate, etc.
ANSWER: Ethical reporting codes have been added to the manuscript, as suggested by the reviewer.
2) Motivations and new contributions/originality should be mentioned in introduction.
ANSWER: The introduction has been rephrased. We have better emphasized the motivation and background, highlighting the real contributions of this work: 1) look for muscular forearm areas that are significantly affected by HOA in EMG terms; 2) study if the affected EMG characteristics can be used as predictors to detect HOA in an early stage by using different combinations of them in discriminant analyses
3) Section 2.1: Experiments should be detailed.
ANSWER: Section 2.1 has been improved and a new figure has been added in order to better detail the experimentation.
4) Quality of figures: Numbers in pictures are quite small.
ANSWER: Captions of tables and figures have been improved to make them clearer.
Round 2
Reviewer 2 Report
The authors reacted properly to my pointed issues.
Reviewer 4 Report
Accepted in current form